# PatchFormer: A neural architecture for self-supervised representation learning on images

## Abstract

Learning rich representations from predictive learning without labels has been a longstanding challenge in the field of machine learning. Generative pre-training has so far not been as successful as contrastive methods in modeling representations of raw images. In this paper, we propose a neural architecture for self-supervised representation learning on raw images called the PatchFormer which learns to model spatial dependencies across patches in a raw image. Our method learns to model the conditional probability distribution of missing patches given the context of surrounding patches. We evaluate the utility of the learned representations by fine-tuning the pre-trained model on low data-regime classification tasks. Specifically, we benchmark our model on semi-supervised ImageNet classification which has become a popular benchmark recently for semi-supervised and self-supervised learning methods. Our model is able to achieve 30.3% and 65.5% top-1 accuracies when trained only using 1% and 10% of the labels on ImageNet showing the promise for generative pre-training methods.

## 1 Introduction

Deep neural networks are capable of learning rich abstract representations from raw high dimensional data in an end-to-end fashion (LeCun et al., 2015). A big weakness of these neural networks is the reliance on abundant labeled datasets. Self-supervised and unsupervised representation learning approaches have been proposed to address this problem (Bengio et al., 2007). It is still an open problem in the field to figure out how to take advantage of large unlabeled datasets, use them for learning rich representations and improving the data-efficiency of supervised learning systems.

A classic example of successful unsupervised learning of rich representations is word2vec (Mikolov et al., 2013) where the authors showed that distributed vector representations of words could be learned by contrastively predicting the neighboring words given surrounding words. The shift from word embeddings to sequence embeddings in recent times began when (Dai & Le, 2015) showed that pre-trained sequence to sequence autoencoders on text corpora could be useful for a number of downstream tasks such as text classification and sentiment analysis. Followed by this, it was shown in (Peters et al., 2018) that language modeling is useful in providing deep contextual sentence embeddings that could be fine-tuned on a number of natural language understanding tasks. (Howard & Ruder, 2018) is another example of such a success. In more recent times, the transformer (Vaswani et al., 2017) has emerged as a powerful architecture to model complex dependencies across a long sequence using global self-attention. OpenAI Generative Pre-Training (GPT) (Radford et al., 2018) showed that training large Transformer models on BooksCorpus could lead to rich and useful representations that could be fine-tuned on a variety of downstream tasks covering language understanding, commonsense reasoning and question-answering. The biggest success in unsupervised pre-training was achieved by BERT (Devlin et al., 2018) where the assumption for using causal language modeling was pointed out as unnecessary and it was shown that training deep transformers in a bi-directional fashion to perform the objective of masked language modeling and next sentence prediction could lead to rich and useful representations covering a wide span of natural language understanding downstream tasks.

Therefore, it is useful to address the following question: How do we translate the successes of masked language modeling and deep transformers to images? Unlike language which is a layer of abstraction to be able to understand the world and communicate thoughts, images are raw sensory observations. It is therefore much harder to model the relationship across pixels both spatially and temporally simply because the dimensionality is much higher.

Let's first look at the question of whether generative pre-training is well suited for images or not. There is a belief that generative approaches are more suited to abstract inputs such as language wordpieces but not for less abstract entities like pixels or audio waveform bits (van den Oord et al., 2018; Hjelm et al., 2018; Bachman et al., 2019; Trinh et al., 2019). While it may as well turn out to be true, it is useful to investigate how far we could push generative approaches for pre-training even on domains they are not well suited for, such as images.

A successful example of such an approach is the adversarial method BiGAN (Donahue et al., 2016; Donahue & Simonyan, 2019). While BiGAN (and BigBiGAN) are meant for learning useful high-level representations of raw images, they still retain the generative modeling aspect of unsupervised learning by learning to jointly model an encoder and a generator using the generative adversarial loss.

On the other hand, there has been incredible progress in recent years in generative modeling of raw pixels and audio waveforms using maximum likelihood. Beginning with (Oord et al., 2016b), we have seen successes in generating diverse images by modeling the conditional distribution of pixels given context of neighboring pixels. WaveNet (Oord et al., 2016a) is an example of successful deployment of such techniques for modeling the distribution of raw audio waveforms when conditioned on text. (Kalchbrenner et al., 2017) adopt a similar technique for generating future frames of a video conditioned on the past. More recently, (Child et al., 2019) have pushed on using strided self-attention to achieve high-quality unconditional samples of ImageNet building upon successes of (Parmar et al., 2018) and (Menick & Kalchbrenner, 2018).

Therefore, it is very reasonable to ask ourselves the following question: If generative models can work on such high dimensional data, is it necessarily the case that they would be ill-suited from a representation learning perspective? If no, how do we leverage these successes for representation learning? Further, how do we take inspiration from the big representation learning successes in natural language processing (Devlin et al., 2018) and the generative modeling successes for images and audio and design a representation learning approach for images?

As far as representation learning on images goes, the state-of-the-art systems at the moment are contrastive methods. Specifically, Contrastive Predictive Coding (CPC) (van den Oord et al., 2018) which learns to contrastively predict the future given the past by sampling negatives across and between sequences has been shown to be a universally powerful representation learning approach for multiple modalities (audio, images, text, control). (Hénaff et al., 2019) and (Bachman et al., 2019) achieve impressive linear classifier probe metrics for their representations that were trained contrastively to maximize mutual information across views and space. (Hénaff et al., 2019) also show that these representations could be used for downstream tasks such as semi-supervised image classification in the low-data regime going on to record impressive results in the 1% and 10% ImageNet classification.

While such impressive results have been shown using the contrastive methods, methods of such quality for generative approaches are ye to be shown on images. Secondly, CPC and related methods adopt convolutional architectures for learning the representations. We believe it is worth the research effort to investigate architectures that incorporate self-attention so that we could translate language domain's success to other domains. Stand-Alone Self-Attention (Ramachandran et al., 2019) has shown that self-attentive architectures could be designed to match convolutional architectures on image classification and object detection. Such a result is promising in the sense that we now know that self-attentive architectures are not a limiting factor for downstream classification performance.

In this paper, we attempt to inspire from a few key engineering deicisons that have benefitted the various successful approaches discussed above to motivate our design of a generative pre-training method for images.

1. **Predicting subscales and low-bit depth for pixels:** (Menick & Kalchbrenner, 2018) showed that modeling pixels by sequentially modeling the subscales and low-bit depth

versions of the raw image is extremely useful. (Oord et al., 2016a) also attempted to initially model 8-bit audio rather than 16-bit. Therefore, it makes sense to model the only the most significant few bits while attempting to decode pixels for representation learning. Higher order bits are more relevant for texture and finer-details and may not be crucial for representation learning performance.

2. **Use of self-attention for aggregating global context:** Self-Attention (Vaswani et al., 2017) is an extremely powerful approach for aggregating global contextual representations across large sequences. The adoption of self-attention for images began with (Wang et al., 2018) who used non-local layers for activity recognition. (Zhang et al., 2018) and (Brock et al., 2018) exploit non-local layers for high-fidelity image generation. (Razavi et al., 2019) has also shown that self-attention can be used to good effect for modeling distribution of latents for likelihood-based image generation while (Parmar et al., 2018; Menick & Kalchbrenner, 2018; Child et al., 2019) are examples for self-attentive density models.

3. **Learning spatial dependencies across patches:** CPC learns to spatially predict neighboring patches given context of surrounding patches. Image Transformers (Parmar et al., 2018) adopts self-attention that takes into account local as well as global dependencies behaving like a patch-based generative model. (Menick & Kalchbrenner, 2018) exploit modeling spatial PixelCNNs over subscales for global image dependencies. (Trinh et al., 2019) attempt to modify CPC for image representation learning by using the patch-based data extraction and modeling dependencies in a BERT-like fashion using self-attention.

Our key contributions are as follows:

1. We propose a new architecture, PatchFormer, for modeling bi-directional dependencies across patches. Our architecture learning to decode missing patches in an image by extracting representations of the given patches, using attention-pooling to aggregate the context, and decode the *low-bit grayscale sub-sampled versions* of the missing patches. Specifically, we decode only the 2-bit grayscale version of the missing patch.

2. We show that our model could be pre-trained on the unsupervised objective of decoding missing patches and fine-tuned on downstream low-data regime classification tasks.

3. We achieve somewhat competitive downstream ImageNet classification results with CPC (Hénaff et al., 2019) and are surprisingly even better than the other contrastive approach for semi-supervised downstream classification, Selfie (Trinh et al., 2019) in spite of adopting a generative approach.

## 2    ARCHITECTURE

### 2.1    PRE-TRAINING

Our patch-extraction setup is described in Figure 1. Our input images are 224x224x3. We extract 16x16 patches from this global image with a stride of 16x16. This results in a grid of 14x14 patches. Among the 296 patches, we spatially mask 60% of the patches (118) and use the remaining 78 patches. Our masks are designed with a 7x7 grid first (with the same masking ratio) and then upsampled to 14x14 with nearest-neighbor upsampling. This is to ensure that our masks are contiguous and blocky to make the prediction task harder.

We then use a Transformer to spatially associate the convolutional features extracted from the non-masked patches. Our self-attention mechanism flattens the grid and adds learned position position embeddings to each. To regularize the model, we use factorized position embeddings (similar to (Trinh et al., 2019)) for the X and Y coordinates. We use 10 layers of masked-self-attention with 1024 dimensional embeddings and 16 attention heads. Our pointwise MLP uses the GeLU non-linearity and a widening layer of 4x similar to BERT.

We then use the extracted attention-pooled context vectors to decode the missing patches. We use a regular residual upsampling decoder to decode 8x8 versions of the missing patches originally of shape 16x16. The ground truth for the missing patches comes from the 2-bit gray-scale centre crop of these patches. We also additionally add soft-label noise to our cross entropy loss (0.01).

Our model is trained with AdamWeightDecay Optimizer using a learning rate of 2e-4 and weight-decay coefficient of 0.001 with a batch size of 128 on a v3-32 cloud TPU over 300,000 update steps.

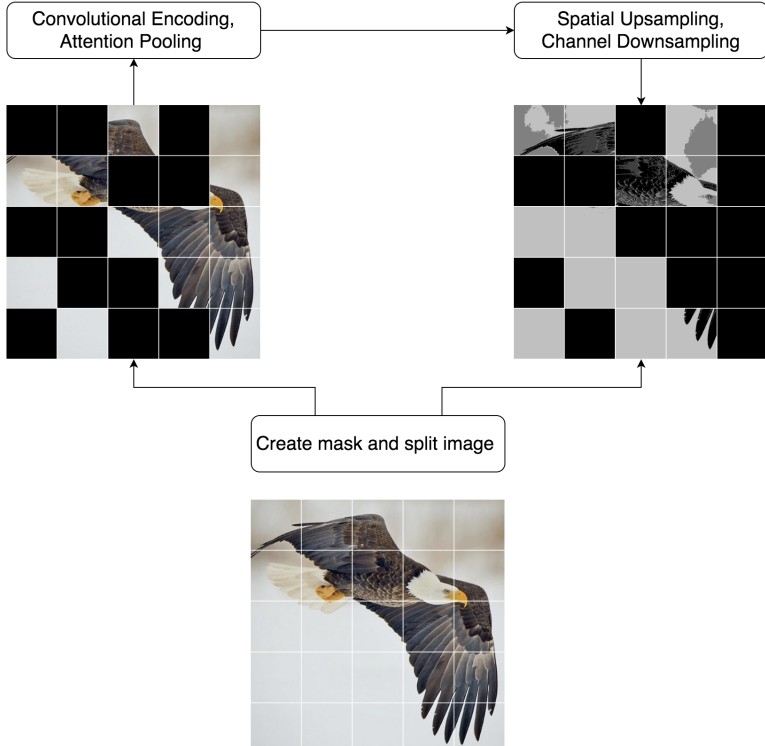

Figure 1: PatchFormer Architecture: We extract a grid of non-overlapping patches from the raw image. We then pick random crops of unmasked patches for spatial jittering. This is done to ensure the model does not cheat similar to (van den Oord et al., 2018). We then apply random data-augmentations to these patches such as Inception Noise (random brightness, hue, saturation and contrast) as well as arbitrarily flip the patches 20% of the times. We also randomly convert the patches to their low-bit depth versions with the depth sampled randomly from 2 to 8 and apply random grayscale blending with 20% probability. The pixels are then scaled to [-1, 1] after applying these augmentations. A convolutional neural network (ResNet-41 - a customized version of the first 3 stacks of ResNet-50) is applied to extract the embeddings for these patches. A deep self-attentive transformer then globally associates the patches across space (with added factorized position encodings) before decoding the missing patches. We decode subsampled and low-bit grayscale version of the missing patches to make the prediction task simpler and more suited for representation learning.

## 2.2 FINE-TUNING

We remove the decoder and extract the attention-pooled embedding of patches for classification (a residual layer operates on top of the attention pooled embedding). To ensure there is no mismatch between training and fine-tuning times (because of masking while pre-training), we randomly mask out 10% of the patches even while fine-tuning. While this does hurt the model in terms of good performance, it also helps by regularizing the model. We fine-tune the model using the same optimizer as used for pre-training with batch sizes in {16, 32, 64, 128, 256} and learning rates in {3e-5, 2e-4, 4e-4, 1e-3}. We also use soft-labels while fine-tuning to add some regularization to the model. Additionally, our model also employs dropout noise at the end to prevent overfitting on the low-data regime classification. The residual block we use on top of the attention-pooling is the same as what we use for our convolutional encoder described in Figure 2.

## 3 RESULTS

We pre-train using the 1.27 million unlabeled ImageNet training dataset. We then fine-tune for image classification on 1%, 10% and 20% of the dataset and report the accuracies (top-1 and top-5) on the validation set of 50000 images for each of these subsets.

Table 1: Top-1 Accuracies

| Model | 1% ImageNet | 10% ImageNet | 20% ImageNet |
|---|---|---|---|
| CPC++ | Unknown | Unknown | Unknown |
| Selfie | Unkown | 61.9 | 67.1 |
| Ours | **30.3** | **65.5** | **70.2** |

Table 2: Top-5 Accuracies

| Model | 1% ImageNet | 10% ImageNet | 20% ImageNet |
|---|---|---|---|
| CPC++ | 64.03 | 84.88 | 88.5 (based on plot) |
| Selfie | Unknown | Unknown | Unknown |
| Ours | **52.0** | **82.3** | **87.6** |

## 4 CONCLUSION

We have proposed a new architecture for generative pre-training on images called the PatchFormer. We highlighted the key tricks to making our model learn useful representations for downstream classification tasks in spite of decoding pixels. We have shown that we are competitive with state-of-the-art contrastive pre-training methods such as CPC on the low data-regime ImageNet classification benchmark.

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

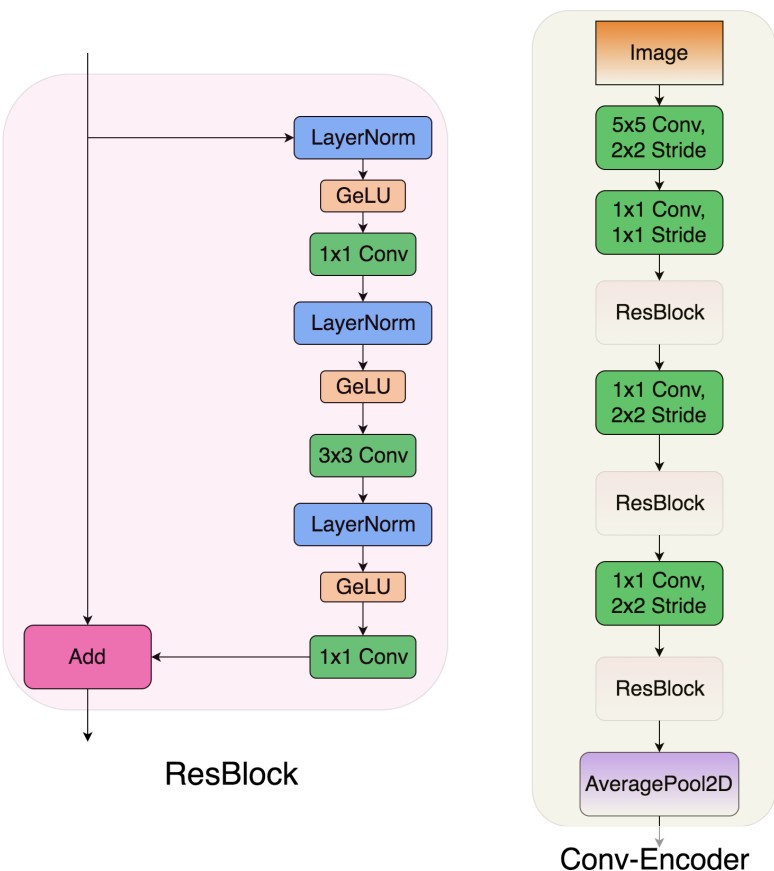

Figure 2: Convolutional Encoder used for patch embeddings

