# OpenReview forum: "PatchFormer: A neural architecture for self-supervised representation learning on images"
_ICLR.cc/2020/Conference — Reject_

### Official Review · AnonReviewer3 · 2019-10-22
**Official Blind Review #3**

**Rating:** 1

**Review:**

The motivation of this paper is to use the idea of Transformer-based NLP models in image data, which is appreciated. However, this seems to be a far unfinished paper. The introduction part is well written. But, the method is not well described.  It is very unclear how exactly the model is built. Moreover, the network structure in Figure 2 is not explained.  The experimental part is very brief, and unconvincing. Much more investigations and comparisons are needed.

Minors:
deicisons?
model the only the most significant few bits -> model only the most significant few bits
position position embedding -> position embedding


**Experience Assessment:**

I have read many papers in this area.

**Review Assessment: Checking Correctness Of Derivations And Theory:**

I carefully checked the derivations and theory.

**Review Assessment: Checking Correctness Of Experiments:**

I carefully checked the experiments.

**Review Assessment: Thoroughness In Paper Reading:**

I read the paper thoroughly.

---

### Official Review · AnonReviewer2 · 2019-10-25
**Official Blind Review #2**

**Rating:** 1

**Review:**

This paper attempts unsupervised representation learning, via a patch prediction task on ImageNet. The paper is sparse on details, but the method appears to be: (1) split the image into non-overlapping visible and masked patches, (2) from features extracted from the visible patches, predict the masked patches. Rather than predict RGB, they choose to predict 2-bit grayscale images. Also, rather than use the full patches, they use random crops of the input ones, and a center crop of the output ones.

The paper seems to be an early draft of something bigger, submitted with the hope of getting some feedback. The method description is mostly composed of tiny details, such as the number and sizes of the patches; I recommend rewriting this to focus on the big idea first, and pack the details into another sub section like "Implementation Details". The paper barely includes any evaluation. Also, the method does not appear to be very novel: I recommend the authors look at and compare against "Unsupervised Visual Representation Learning by Context Prediction" (ICCV 2015), which is conceptually very similar.

The evaluation right now is not good. "Unknown" is not a valid point of comparison. I understand the code for CPC++ might not be released yet, but the authors could at least implement their best approximation of it, and also find older works (which CPC compared against in their paper), to fill out the results and make a convincing argument.

In Table 2, the proposed model performs worse than CPC++, yet its values are bolded anyway. Please only put the best-performing result in bold.

**Experience Assessment:**

I have read many papers in this area.

**Review Assessment: Checking Correctness Of Derivations And Theory:**

I assessed the sensibility of the derivations and theory.

**Review Assessment: Checking Correctness Of Experiments:**

I assessed the sensibility of the experiments.

**Review Assessment: Thoroughness In Paper Reading:**

I read the paper at least twice and used my best judgement in assessing the paper.

---

### Official Review · AnonReviewer1 · 2019-10-26
**Official Blind Review #1**

**Rating:** 1

**Review:**

Contributions:
The paper aims to develop generative pre-training method for learning representations of images. Although representation learning for images has been widely investigated, the present work distinguishes itself by a combination of the following:
a) building on the use of transformers as a series of layers after initial convolutional layers;
b) using self attention for aggregating context;
c) learning spatial dependencies across patches;
d) training on the task of predicting two bit gray scale version of randomly masked patches in an image

Results:
Limited experiments aim to compare against the CPC (Hefnaf et al 2019) and Selfie (Trinh et al 2019) algorithms both of which are contrastive unlike the generative approach adopted in the paper. After pre-training on unlabeled imageNet datasets the proposed approach is competitive with these algorithms with roughly similar results.

Evaluation/Suggestions:
Overall the paper combines ideas from several previous works in ways that are not sufficiently novel in the opinion of this reviewer and the experiments are very limited to the imageNet dataset with 1%, 10% and 20% of labels provided to downstream classification modeling, and evaluated on top-1 and top-5 accuracies. The paper could improve on its experimental evaluation bycomparing on multiple datasets, showing error bars when averaging across multiple samplings (eg for getting the 1% label set from the entire imageNet dataset) and also comparing with other approaches even when they dont directly aim to learn representation from unlabeled data (eg Image Transformers by Parmar et al). In addition the description is very high level and does not provide enough details for experimental reproducibility. For example the reviewer had to actually guess at some of specifics of the overall end-to-end architecture since it was not fully described precisely eg in a diagram. It would be relatively easy (but important) to provide suffcient detail for reproducibility

**Experience Assessment:**

I have published one or two papers in this area.

**Review Assessment: Checking Correctness Of Derivations And Theory:**

N/A

**Review Assessment: Checking Correctness Of Experiments:**

I carefully checked the experiments.

**Review Assessment: Thoroughness In Paper Reading:**

I read the paper thoroughly.

---

### Decision · Program_Chairs · 2019-12-19

**Decision:**

Reject

**Comment:**

The paper presents a generative approach to learn an image representation along a self-supervised scheme.

The reviews state that the paper is premature for publication at ICLR 2020 for the following reasons:
* the paper is unfinished (Rev#3); in particular the description of the approach is hardly reproducible (Rev#1);
* the evaluation is limited to ImageNet and needs be strenghtened (all reviewers)
* the novelty needs be better explained (Rev#1).
It might be interesting to discuss the approach w.r.t. "Unsupervised Learning of Visual Representations by Solving Jigsaw Puzzles", Noroozi and Favaro.

I recommend the authors to rewrite and better structure the paper (claim, state of the art, high level overview of the approach, experimental setting, discussion of the results, discussion about the novelty and limitations of the approach).